# Impact of Real-World Outpatient Cancer Rehabilitation Services on Health-Related Quality of Life of Cancer Survivors across 12 Diagnosis Types in the United States

**DOI:** 10.3390/cancers16101927

**Published:** 2024-05-18

**Authors:** Mackenzi Pergolotti, Kelley C. Wood, Tiffany D. Kendig, Stacye Mayo

**Affiliations:** 1ReVital Cancer Rehabilitation, Select Medical, Mechanicsburg, PA 17055, USA; kecwood@selectmedical.com (K.C.W.); tkendig@selectmedical.com (T.D.K.); smayo@selectmedical.com (S.M.); 2Department of Occupational Therapy, University of North Carolina at Chapel Hill, Chapel Hill, NC 27599, USA

**Keywords:** cancer rehabilitation, cancer, health-related quality of life, cancer survivors, patient-reported outcomes, occupational therapy, physical therapy

## Abstract

**Simple Summary:**

This is the first study to examine the impact of specialized community-based outpatient physical and occupational therapy on the health-related quality of life of cancer survivors with various cancers. Survivors who completed PT/OT reported significant improvements in physical and mental health, physical function, and social participation. Although more research is needed, these findings suggest cancer rehabilitation is an important strategy to meet HRQOL needs and lends support for inclusion as a standard of care.

**Abstract:**

Compared to adults without cancer, cancer survivors report poorer health-related quality of life (HRQOL), which is associated with negative treatment outcomes and increased healthcare use. Cancer-specialized physical and occupational therapy (PT/OT) could optimize HRQOL; however, the impact among survivors with non-breast malignancies is unknown. This retrospective (2020–2022), observational, study of medical record data of 12 cancer types, examined pre/post-HRQOL among cancer survivors who completed PT/OT. PROMIS^®^ HRQOL measures: Global Health (physical [GPH] and mental [GMH]), Physical Function (PF), and Ability to Participate in Social Roles and Activities (SRA) were evaluated using linear mixed effect models by cancer type, then compared to the minimal important change (MIC, 2 points). Survivors were 65.44 ± 12.84 years old (range: 19–91), male (54%), with a median of 12 visits. Improvements in GPH were significant (*p* < 0.05) for all cancer types and all achieved MIC. Improvements in GMH were significant for 11/12 cancer types and 8/12 achieved MIC. Improvements in PF were significant for all cancer types and all achieved the MIC. Improvements in SRA were significant for all cancer types and all groups achieved the MIC. We observed statistically and clinically significant improvements in HRQOL domains for each of the 12 cancer types evaluated.

## 1. Introduction

Compared to adults without cancer, adults living with or beyond cancer (cancer survivors) experience poorer health-related quality of life (HRQOL) [1,2]. Of the 18 million survivors living in the United States of America (USA), up to 75% report unmet needs in one or more domains of HRQOL, including physical or mental health, physical function, and social participation [3,4,5,6,7,8,9,10,11]. The risk for poorer cancer outcomes increases when domains of HRQOL are left unaddressed, including morbidity and mortality [12,13,14], poor treatment tolerance [15,16], increased caregiver burden [17,18], and increased use of long-term care and unplanned hospitalizations [19]. Declines in HRQOL are a population-based issue for cancer survivors and our healthcare system that will only grow with the significant increase in older adults with cancer [20], and meeting survivors’ HRQOL needs is critical to achieve better patient health and experience while optimizing costs. Although the cancer care team as a whole is accountable for addressing survivors’ HRQOL needs, the ability to identify these needs and provide targeted intervention is a long-standing challenge in oncology care [21,22].

Proposed solutions to this population-based issue include the integration of supportive services within the typical oncology workflow, such as outpatient cancer rehabilitation services (defined here as cancer-specialized physical and occupational therapy services (PT/OT)) [23]. PT/OT are skilled services covered by most insurances with the goal of reducing the negative impact of treatment-related side effects on daily life, improving functional independence, participation in life roles, and HRQOL [24]. Meta-analyses of controlled research studies demonstrate the efficacy of PT/OT interventions in alleviating treatment-related side effects (pain, lymphedema, fibrosis, atrophy, etc.) and optimizing functional independence (ability to complete self-care, daily or social activities, return to work, etc.) [25,26,27,28]. In addition, growing evidence shows participation in cancer rehabilitation interventions may result in cost savings to survivors and the healthcare system [29,30,31]. However, the limited generalizability of existing evidence is a barrier to understanding the population-level HRQOL benefits of community-based outpatient PT/OT services and may potentially limit access to care [23,32,33]. This is especially true for non-breast malignancies and medically complex populations, many of whom are excluded from medical clinical trials but experience high rates of functional and participation restrictions amenable to rehabilitation [1,23,26].

A systematic evaluation of practice-based evidence obtained from real-world, nationally representative outpatient PT/OT care is needed to understand the degree to which participation in these services may improve HRQOL for cancer survivors of multiple cancer types. To address this research–practice gap and public health needs, we performed a retrospective, pre/post, observational study of rehabilitation medical records, to better understand the impact of real-world PT/OT services on HRQOL outcomes for cases with non-breast malignancies (breast malignancies reported elsewhere) [34]. We hypothesized that we would observe significant improvement in HRQOL outcomes among individuals within each cancer type from initial evaluation to discharge. 

Primary outcomes include global physical and mental health, and secondary outcomes include physical function and ability to participate in social roles and activities. All outcomes were collected during routine PT/OT services by licensed therapists with cancer-specific training, using Patient Reported Outcome Measurement Information System (PROMIS^®^) short forms.

## 2. Materials and Methods

This was a retrospective study of rehabilitation medical record data from patients who completed outpatient cancer PT/OT (N = 1415) over a two-year period (2020–2022) by a large rehabilitation network (ReVital Cancer Rehabilitation Program, Select Medical). The services were provided in 604 clinics across 21 states in the USA. Rehabilitation interventions were provided by cancer-specialized PT and/or OT and billed through insurance for skilled therapy as per typical practice. Rehabilitation interventions, visit frequency, and duration were determined by the evaluating therapist based on the clinical evaluation and the survivor’s individual needs and goals. We report the study methods following the Professional Society for Health Economics and Outcomes Research (ISPOR) checklist for retrospective studies [35].

### 2.1. Case Identification and Data Extraction

Rehabilitation cases that met the following criteria were included in the study: history of a non-breast malignancy (identified from ICD-10 codes), 18 years of age or older, discharged from PT/OT during the study period (January 2020 and September 2022), attended three or more PT/OT visits, and had the primary outcome measure available at initial evaluation (pre) and discharge (post). An honest broker used these criteria to identify eligible cases in the EMR (N = 1415) and create a de-identified dataset for analysis. Available case characteristics included age, sex, race, USA region, payer-type rehabilitation service type (PT/OT), ICD-10 codes, number of visits, and length of care (weeks). Cancer type was determined from ICD-10 codes. 

### 2.2. PROMIS Measures and Scoring

Developed and validated by the National Institutes of Health (NIH), PROMIS measures are recommended to evaluate HRQOL in general and clinical populations due to high-precision and low-response burden [36,37,38,39,40]. Each PROMIS measure is scored using a T-score scale, where T-score of 50 represents the general population mean in the USA (standard deviation: 10) [37,38]. Higher T-scores indicate more of the domain being measured (i.e., global physical and mental health, physical function, ability to participate in social roles or activities) and have shown to be sensitive and specific to capture improvement or decline in HRQOL during or following cancer treatment [41]. The established minimal important change (MIC) is two to six points on the T-score scale [42]. We used 2 points as the threshold for clinically significant change in this study. Three PROMIS measures (with four domain scores) were recommended and routinely used by treating therapists of the national program due to high feasibility, sensitivity, and specificity for evaluating cancer survivors’ HRQOL [2,43,44,45,46,47]; these included the 10-item Global Health measure (scored with two domains, Global Physical Health [GPH] and Global Mental Health [GMH]) [48], the four-item Physical Function (PF) measure, and the four-item Ability to Participate in Social Roles and Activities (SRA) measure. We scored each measure into the appropriate T-score using established guidelines and categorized baseline T-scores using established cutoff points (e.g., GPH T-score below 42 categorized as fair-to-poor) [40].

### 2.3. Covariates

All covariates were extracted from the rehabilitation medical record and chosen based on availability and potential impact on the primary outcome. Covariates included in the model were as follows: age, sex, rehabilitation service type, visits, and length of care. 

### 2.4. Statistical Analysis

We evaluated case characteristics descriptively using frequencies or measures of central tendency (mean ± SD or median, IQR), including the prevalence of impairment in each HRQOL domain at baseline as determined by the established PROMIS cut points (Figure 1) [40]. To identify cancer types with above-average levels of HRQOL impairment at baseline, we performed one-sample binomial testing (Figure 1). To examine change pre- to post-rehabilitation, in HRQOL measures among each cancer type, we used linear mixed effect models. Covariates included in each model are described in Table 1 and include age, sex, rehabilitation service type, number of visits, and length of care. From each model, we report the estimated marginal (EM) means and standard error (SE) at each timepoint (pre/post), EM mean change between timepoints, and the p-value for timepoint (main effect). Because a large portion of each group completed PT (93%) as compared to OT services, we adjusted all EM means to the rehabilitation service-type sample margins. Statistical analyses were completed using IBM SPSS Statistics (Version 28) or R (R version 4.0.1) [49]. Alpha was set to 0.05 for all hypothesis tests. 

## 3. Results

On average, cases were 65.45 ± 12.84 years old, male (54%), and completed a median of 12 visits (IQR: 8.0 to 19.0) over 28 weeks (IQR: 9.1 to 80.0). Most completed PT (92.9%) versus OT (7.1%). Common cancer types included head and neck (*n* = 257, 18%), lung (*n* = 169, 12%), gynecologic (*n* = 158, 11%), and prostate (*n* = 146, 10%). All characteristics and the prevalence of each cancer type are reported in Table 1. Overall prevalence of HRQOL impairment at baseline was 54% in GPH, 20% in GMH, 78% in PF, and 60% in SRA (Figure 1). A significantly greater proportion of impairment was reported by cases with gastrointestinal (67.5%, *p* = 0.009), lung (66.3%, *p* < 0.001), and gynecologic (62.0%, *p* = 0.026) cancers for GPH; by cases with pancreatic (30%, *p* = 0.026) and multiple myeloma (27%, *p* = 0.042) for GMH; by cases with pancreatic (90.9%, *p* = 0.009), lung (90.8%, *p* < 0.001), brain/spinal cord/CNS (90.6%, *p* = 0.021), and gastrointestinal cancers (88.7%, *p* = 0.021) for PF; and by cases with pancreatic (80.3%, *p* < 0.001), gastrointestinal (72.9%, *p* = 0.019), lung (69.7%, *p* = 0.009), lymphoma (69.6%, *p* = 0.039), and leukemia (64.5%, *p* < 0.001) cancers for SRA. A significantly lower proportion of impairment was observed in head and neck for GPH (36.2%, *p* < 0.001), GMH (14.0%, *p* < 0.001), and PF (50.6%, *p* < 0.001). 

**Figure 1 cancers-16-01927-f001:**
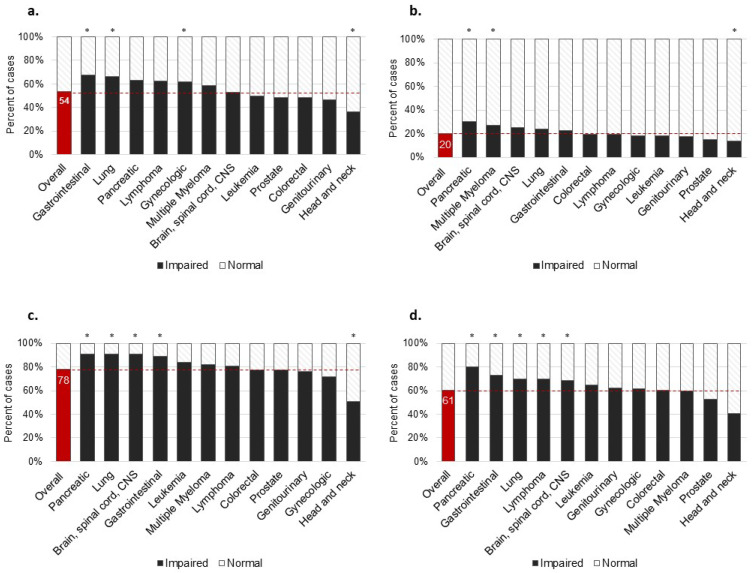
Percentage of cases reporting impairments in HRQOL domains at baseline, overall, and by cancer type. Abbreviations HRQOL, health-related quality of life. HRQOL impairment was determined using established cut points for each PROMIS^®^ measure: (**a**) global physical health (T-score ≤ 41), (**b**) global mental health (T-score ≤ 39), (**c**) physical function (T-score ≤ 45), and (**d**) ability to participate in social roles and activities (T-score ≤ 45) [40]. * The within-group percentage of impairment was significantly different than the percentage of impairment observed overall, *p* < 0.05.

From initial evaluation to discharge, HRQOL improved significantly overall and among each cancer type (all *p* < 0.05, Table 2, Table 3, Table 4 and Table 5). Improvements in GPH were significant (*p* < 0.05) for all cancer types (range: 2.93 [colorectal] to 5.15 [leukemia]), and all achieved MIC (Table 2). Improvements in GMH were significant for 11/12 cancer types (range: 1.83 [lung, prostate] to 3.24 [brain/CNS]), and 8/12 achieved MIC (Table 3). Of the cancer types that did not achieve the MIC for GMH, average improvements trended in that direction, ranging from 1.83 to 1.94. Improvements in PF were significant for all cancer types (range: 2.66 [head and neck] to 4.61 [leukemia]), and all achieved the MIC (Table 4). Improvements in SRA were significant for all cancer types (range: 2.16 [prostate] to 5.53 [leukemia]), and all groups achieved the MIC (Table 5). 

## 4. Discussion

In this study of over 1400 adults, across 12 different cancer types, who completed outpatient cancer rehabilitation, we observed statistically and clinically significant improvements in each domain of HRQOL measured (physical and mental health, physical functioning, and social participation). This is the first study to evaluate the impact of cancer PT/OT on HRQOL in a large, nationally representative sample, participating in real-world outpatient community-based services. Furthermore, this is the first study to examine rehabilitation outcomes in 12 unique cancer types, many of which are understudied. In doing so, we provide a pragmatic and generalizable understanding of how patients within several different cancer populations may present on evaluation to rehabilitation and their potential for improvement. Coupled with existing research demonstrating the high prevalence of unmet HRQOL needs within the growing population of cancer survivors and the potential value of rehabilitation services in terms of patient outcomes [26], satisfaction [50], and cost-effectiveness [29,31], the findings of this study suggest that widespread participation in real-world cancer rehabilitation services could enhance HRQOL for cancer survivors at the population level and add value to oncology care. 

In this study, we corroborate and build upon existing research and the findings of Feliciano and colleagues, who reported that compared to peers without cancer, those with cancer experience poorer HRQOL. In their recent prospective cohort study, the authors [1] reported women with breast, colorectal, endometrial, and lung cancer experienced a significant functional decline after diagnosis and, generally, did not recover to their prior level of functioning, nor to the level of women without cancer 10 years. In this study, we found similar findings at initial evaluation. Survivors reported at evaluation for outpatient PT/OT, their HRQOL scores were one to two standard deviations below the USA average (i.e., PROMIS T-score: 50). Using established PROMIS T-score cut-off values [40], we observed that 20–54% of survivors reported fair-to-poor global mental or physical health, and 60–78% reported mild-to-severe physical functioning and social participation needs. Recognizing that specific HRQOL needs and trajectory differ by cancer type [1,2], we also found clinically meaningful differences across cancer types on evaluation for rehabilitation. For example, survivors with pancreatic cancer had the lowest average scores for GPH, GMH, and SRA, while patients with lung and brain/spinal cord/nervous system cancers reported the lowest PF scores. 

Although at baseline, average group scores were below USA norms for all HRQOL domains, average scores among several cancer types met or exceeded USA norms when discharged from cancer PT/OT. GMH scores for 4/13 types (head and neck, genitourinary, leukemia, prostate) and SRA scores for 2/13 cancer types (head and neck, leukemia) met or exceeded USA norms. The magnitude of improvement in HRQOL outcomes observed in this study was similar to or greater than the findings of previous meta-analyses and our previous evaluations of practice-based evidence [26,51,52,53]. We also observed differences in the magnitude of change across domains (see Table 2, Table 3, Table 4 and Table 5). For example, the leukemia (GPH and SRA), lung (PF), and genitourinary (GMH) groups experienced the greatest magnitude of improvement across the HRQOL domains. These diagnosis-specific differences are important for recognition and screening, triggering a referral, and during rehabilitation evaluation and intervention. Taken together, these results demonstrate that participation in rehabilitation elicits important improvements in some domains of HRQOL. Notably, improvements associated with cancer PT/OT allowed survivors to achieve HRQOL that was, at best, similar to individuals without cancer or, at least, greater than that observed for individuals in other studies who may not have had access to this supportive care service [54,55]. Because poor PROMIS HRQOL outcome scores have been closely associated with morbidity and mortality after cancer diagnosis [56,57], these results suggest that rehabilitation may also have a positive effect on these outcomes; however, more research is needed.

Patient-reported improvements in mental health (GMH) in this study were especially noteworthy. Mental health support is critical considering that up to 65% of individuals with cancer report high levels of distress and depressive or anxious symptoms [17,58,59,60]. In some cases, high levels of distress are related to unplanned healthcare use [61] and decreased adherence to cancer treatments [62]. However, statistically significant (12/13 groups) and clinically significant improvement in GMH was observed across all the 12 cancer-type groups. It is plausible that this may be due to the levels of distress and decreased mental health influenced by a change in one’s physical ability or level of independence [17,60,63]. Potentially, cancer PT/OT combined with other supportive services such as social work, rehabilitation psychology, and psycho-oncology could have an even greater impact on survivors’ mental health specifically and should be investigated. 

The results of this study add to the literature in three ways: First, analysis of data from a large, nationally representative sample of cancer rehabilitation participants lends to the generalizability of findings to outpatient PT/OT interventions that survivors are likely to encounter in the United States. Cases were treated in 21 U.S. states, and the community-based nature of the PT/OT services provided is representative of where cancer care is increasingly provided in the United States. Therefore, the results may serve as an appropriate viewpoint allowing survivors or providers alike to anticipate the potential benefits of rehabilitation as well as provide a benchmark for comparison for rehabilitation providers regarding the quality of outcomes achieved. Second, this study evaluated the impact of PT/OT across 12 non-breast malignancies, where evidence of interventions for these populations is severely limited, yet higher rates of functional and participation restrictions remain. Third, the use of PROMIS outcomes in this study has two key benefits: (1) ability to easily compare the HRQOL of our sample to the USA general population, and (2) ability to demonstrate the use of cancer PT/OT as a means to health by demonstrating a ‘dotted line’ to other outcomes of interest in oncology, namely, survival, cost, and patient advocacy. 

### 4.1. Clinical and Public Health Implications

Considering that less than five percent of cancer survivors enroll in medical clinical trials [64,65], the use of real-world clinical data, as we used in this study, is critical to understanding the efficacy of PT/OT services in the community. This allows for a pragmatic evaluation of changes in HRQOL when delivered in contexts representing those that a survivor, at least in the USA, is more likely to experience [66]. We were able to evaluate a larger, more representative sample than would typically be enrolled in a clinical trial, moving toward a broader population-level impact of real-world services (i.e., generalizable and sustainable). Similar improvements in health-related quality of life have been observed in other countries and healthcare settings, including the inpatient setting [67,68,69,70]. For example, Licht et al. (2021) used a similar study design to ours to investigate the effects of an inpatient rehabilitation program in Austria [67]. In their study, which included over 4000 inpatients with various diagnosis types, Licht et al. observed significant improvements in global HRQOL scores. However, many patients reported additional needs upon discharge from inpatient rehabilitation [67,69], often leading to subsequent hospitalization [70]. These needs can and should be managed in the outpatient setting with the goals of keeping patients out of hospital and maximizing HRQOL long-term. The results of this study can be used to enhance access to outpatient cancer rehabilitation services by educating oncology clinicians, survivors, policymakers, and payors on the benefits of these services [23].

For oncology providers, improved integration of specialized outpatient PT/OT services into comprehensive cancer care may be the key to addressing the unmet HRQOL needs of survivors. Outpatient cancer rehabilitation is a relatively low-cost service that is generally covered by insurance. In previous studies of individuals with cancer who attend cancer rehabilitation, patient-rated acceptability was high and most agreed they would recommend the service to others because they could recognize the value in attending [50]. For cancer survivors, previous studies have qualitatively described a lack of understanding of cancer rehabilitation as a barrier to participation [32,33]. These data, coupled with other evidence demonstrating the importance of survivors experiencing an improvement in how they feel [50], relate to the importance of evaluating HRQOL as an outcome of rehabilitation. This study can help inform survivors of what they expect from cancer rehabilitation services, providing an incentive to attend. These positive findings may also interest payors, showing that rehabilitation offers a low-cost means to enhance the outcome of high importance: HRQOL. Finally, this study can be used by researchers, rehabilitation clinicians, or administrators to optimize the translation of research evidence to clinical practice by informing outcome selection, research study design, sample sizes for grants, and future program evaluations. 

### 4.2. Limitations and Future Directions

Owing to the nature of real-world data, this study was limited by the variables available within the rehabilitation medical record, which did not include variables such as cancer stage, grade, date of diagnosis, cancer treatments received, cancer treatment status (active or post-treatment), or reasons for referral that may be available in a more comprehensive medical record, these factors may affect the impact of the intervention. In future studies, researchers should include additional cancer diagnosis (e.g., stage, grade, and time since diagnosis), treatment characteristics (i.e., treatment received and timing), and other health behaviors and supports in their analysis to provide additional understanding of the influence of these factors on the benefits of rehabilitation services. The following considerations may also affect the generalizability of these findings to rehabilitation services delivered outside of this large rehabilitation network in the United States or delivered in other countries. The de-identified dataset used for analysis prevented us from being able to identify cases who may have received multidisciplinary (PT and OT) services and to examine potential differences in cases who completed pre/post outcomes as compared to those who did not. In addition, OT utilization in this study was relatively low (7% of all cases). The authors have previously described the unique role and value of OT services for adults living with and beyond cancer [71], including activities of daily living, participation in life situations, mental health, cognitive function, and overall HRQOL. Additional research and advocacy efforts are needed to expand the cancer-specialized rehabilitation workforce beyond PT to meet the needs of survivors and their caretakers and to advocate for the unique role and benefits of each type of service.

In addition, future studies could build on this study and evaluate the cost–benefit to patients (e.g., out-of-pocket costs compared to costs of longer-term disability and loss of work), providers (e.g., in value-based care models looking to optimize outcomes, unplanned hospitalization, long-term care use, and increased patient satisfaction), and the healthcare system, US as well as others, at large (e.g., payors save money by offering PT/OT earlier and more often to beneficiaries). This could add to our understanding of the value that cancer rehabilitation can bring to all key members. 

## 5. Conclusions

In this large retrospective study, we observed statistically and clinically significant improvements in HRQOL domains reported by over 1400 cancer survivors in 12 unique cancer types who received PT/OT. At evaluation, on average, survivors reported their HRQOL below general population norms, suggesting significant needs, including decreased physical function and social participation. At discharge, HRQOL improved to or above the general population norms for several cancer types. This study builds upon previous work describing the need for improved integration of supportive care and the effectiveness of cancer rehabilitation by demonstrating the population-level impacts observed in a national outpatient cancer rehabilitation program. Future studies to determine the best practices to integrate cancer rehabilitation services into existing oncology workflows and define the cost benefits and survival implications of the service are warranted. 

## Figures and Tables

**Table 1 cancers-16-01927-t001:** Characteristics of cancer survivors participating in outpatient cancer PT/OT, N = 1415.

Characteristic	Mean ± SD, N, %
Age (Mean ± SD, range)	65.45 ± 12.84, 19.0–91.0
Sex (*n*, %)	
Female	650, 45.9%
Male	765, 54.1%
Race/ethnicity (*n*, %) a	
White	538, 80.7%
Black/African American	63, 9.4%
Hispanic	38, 5.7%
Asian or Pacific Islander	27, 4.1%
United States region	
South	585, 39.8%
Northeast	385, 26.2%
Southeast	241, 16.4%
Midwest	177, 12.0%
West	82, 5.6%
Insurer type	
Federally funded	879, 62.1%
Private or self-pay	536, 37.89%
Cancer type	
Head and neck	257, 18.2%
Lung	169, 11.9%
Gynecologic	158, 11.2%
Prostate	146, 10.3%
Colorectal	118, 8.3%
Multiple Myeloma	111, 7.6%
Lymphoma	98, 6.9%
Gastrointestinal	83, 5.9%
Genitourinary	75, 5.3%
Pancreatic	70, 4.9%
Leukemia	66, 4.7%
Brain, spinal cord, nervous system	64, 4.5%
Rehabilitation service type (N, %)	
PT	1314, 92.9%
OT	101, 7.1%
Visits completed (Median, IQR)	12.00, 8.00–19.00
Length of care, weeks (Median, IQR)	28.00, 9.14–80.00

a Race, N = 666. Data unavailable for 52.9%.

**Table 2 cancers-16-01927-t002:** Pre- and Post-Rehabilitation PROMIS^®^ Global Physical Health (GPH) T-scores, by Cancer Diagnosis Type.

	N	Initial Evaluation EM Mean, SE	Discharge EM Mean, SE	EM Mean Change, SE	*p*-Value	Achieved MIC
Head and neck	257	44.45, 0.526	48.41, 0.526	3.96, 0.471 †	<0.001 **	61.9%
Lung	169	39.04, 0.598	43.02, 0.598	3.97, 0.596 †	<0.001 **	63.9%
Gynecologic	158	39.54, 0.526	43.90, 0.526	4.36, 0.489 †	<0.001 **	67.7%
Prostate	146	41.64, 0.667	44.88, 0.667	3.24, 0.503 †	<0.001 **	60.3%
Colorectal	118	42.07, 0.728	45.00, 0.728	2.93, 0.587 †	<0.001 **	61.9%
Multiple Myeloma	111	39.18, 0.739	42.58, 0.739	3.40, 0.601 †	0.001 **	64.0%
Lymphoma	98	39.77, 0.831	43.67, 0.831	3.89, 0.645 †	<0.001 **	65.3%
Gastrointestinal	83	39.14, 0.810	42.51, 0.810	3.37, 0.850 †	<0.001 **	65.1%
Genitourinary	75	42.08, 1.04	45.21, 1.04	3.13, 0.872 †	0.001 **	49.3%
Pancreatic	70	38.81, 0.778	41.88, 0.778	3.07, 0.777 †	<0.001 **	61.4%
Leukemia	66	40.25, 0.958	45.39, 0.958	5.15, 0.810 †	<0.001 **	71.2%
Brain, spinal cord, nervous system	64	40.74, 0.967	44.90, 0.967	4.16, 1.02 †	<0.001 **	68.8%

Pre- to post-rehabilitation within-group change, *p* < 0.05. ** Pre- to post-rehabilitation within-group change, *p* < 0.01. † Estimated marginal (EM) mean change exceeded the minimal important change (MIC; 2 points) [42]. EM means adjusted for the following covariates: age, sex, rehabilitation service type (PT or OT), visits, and length of care.

**Table 3 cancers-16-01927-t003:** Pre- and Post-Rehabilitation PROMIS^®^ Global Mental Health (GMH) T-scores, by Cancer Diagnosis Type.

	N	Initial Evaluation EM Mean, SE	Discharge EM Mean, SE	EM Mean Change, SE	*p*-Value	Achieved MIC
Head and neck	257	48.93, 0.563	51.02, 0.563	1.92, 1.12	<0.001 **	52.1%
Lung	169	45.67, 0.650	48.59, 0.650	1.83, 0.580	<0.001 **	55%
Gynecologic	158	46.42, 0.647	48.82, 0.647	2.20, 0.850 †	<0.001 **	51.9%
Prostate	146	48.54, 0.716	50.35, 0.716	1.83, 0.883	0.002 **	56.8%
Colorectal	118	47.45, 0.801	49.28, 0.801	2.40, 0.550 †	0.002 *	46.6%
Multiple Myeloma	111	45.80, 0.770	47.88, 0.770	2.09, 0.497 †	0.001 **	56.8%
Lymphoma	98	46.94, 0.954	48.88, 0.954	2.72, 0.722 †	0.005 **	53.1%
Gastrointestinal	83	45.82, 0.938	47.66, 0.938	2.93, 0.584 †	<0.001 **	50.6%
Genitourinary	75	48.75, 0.996	50.94, 0.996	1.94, 0.683	0.012 *	57.3%
Pancreatic	70	44.76, 1.00	46.91, 1.00	2.07, 0.643 †	0.020 *	48.6%
Leukemia	66	47.71, 1.04	50.43, 1.04	2.15, 0.901 †	<0.001 **	63.6%
Brain, spinal cord, nervous system	64	46.25, 1.09	48.17, 1.09	3.24, 0.503 †	0.091	51.6%

Pre- to post-rehabilitation within-group change, *p* < 0.05. ** Pre- to post-rehabilitation within-group change, *p* < 0.01. † Estimated marginal (EM) mean change exceeded the minimal important change (MIC; 2 points) [42]. EM means adjusted for the following covariates: age, sex, rehabilitation service type (PT or OT), visits, and length of care.

**Table 4 cancers-16-01927-t004:** Pre- and Post-Rehabilitation PROMIS^®^ Physical Function (PF) T-scores, by Cancer Diagnosis Type.

	N	Initial Evaluation EM Mean, SE	Discharge EM Mean, SE	EM Mean Change, SE	*p*-Value	Achieved MIC
Head and neck	257	44.22, 0.579	46.88, 0.579	2.66, 0.465 †	<0.001 **	45.0%
Lung	169	35.20, 0.625	39.13, 0.625	3.93, 0.601 †	<0.001 **	53.9%
Gynecologic	158	37.84, 0.656	41.44, 0.656	4.36, 0.489 †	<0.001 **	55.9%
Prostate	146	39.52, 0.711	41.89, 0.711	2.37, 0.521 †	<0.001 **	43.4%
Colorectal	118	39.53, 0.808	42.37, 0.808	2.84, 0.550 †	<0.001 **	51.4%
Multiple Myeloma	111	37.96, 0.841	40.75, 0.841	2.78, 0.674 †	<0.001 **	49.5%
Lymphoma	98	38.04, 0.776	41.11, 0.776	3.07, 0.755 †	<0.001 **	50.0%
Gastrointestinal	83	36.66, 0.830	39.45, 0.830	2.79, 0.733 †	<0.001 **	47.9%
Genitourinary	75	39.23, 1.07	42.19, 1.07	2.97, 0.883 †	0.001 **	46.5%
Pancreatic	70	36.74, 0.893	40.00, 0.893	3.26, 0.827 †	<0.001 **	51.5%
Leukemia	66	36.97, 1.03	41.59, 1.03	4.61, 0.855 †	<0.001 **	67.2%
Brain, spinal cord, nervous system	64	35.53, 0.967	38.92, 0.967	3.39, 0.960 †	0.001 **	56.6%

Pre- to post-rehabilitation within-group change, *p* < 0.05. ** Pre- to post-rehabilitation within-group change, *p* < 0.01. † Estimated marginal (EM) mean change exceeded the minimal important change (MIC; 2 points) [42]. EM means adjusted for the following covariates: age, sex, rehabilitation service type (PT or OT), visits, and length of care.

**Table 5 cancers-16-01927-t005:** Pre- and Post-Rehabilitation PROMIS^®^ Ability to Participate in Social Roles and Activities (SRA) T-scores, by Cancer Diagnosis Type.

	N	Initial Evaluation EM Mean, SE	Discharge EM Mean, SE	EM Mean Change, SE	*p*-Value	Achieved MIC
Head and neck	257	49.98, 0.628	52.50, 0.628	2.52, 0.625 †	<0.001 **	45.1%
Lung	169	43.24, 0.680	47.21, 0.680	3.97, 0.670 †	<0.001 **	55.9%
Gynecologic	158	44.84, 0.718	47.87, 0.718	3.03, 0.711 †	<0.001 **	53.0%
Prostate	146	47.00, 0.791	49.16, 0.791	2.16, 0.686 †	00.002 **	40.8%
Colorectal	118	45.72, 0.822	48.88, 0.822	3.16, 0.695 †	<0.001 **	49.1%
Multiple Myeloma	111	43.77, 0.817	47.23, 0.817	3.46, 0.766 †	<0.001 **	48.4%
Lymphoma	98	45.05, 0.945	48.26, 0.945	3.21, 0.886 †	<0.001 **	52.2%
Gastrointestinal	83	43.40, 1.03	47.30, 1.03	3.90, 0.850 †	0.001 **	57.1%
Genitourinary	75	45.19, 1.08	48.78, 1.08	3.59, 0.992 †	0.001 **	47.8%
Pancreatic	70	42.08, 0.846	46.38, 0.846	4.31, 0.802 †	<0.001 **	57.6%
Leukemia	66	45.12, 1.18	50.65, 1.18	5.53, 1.02 †	<0.001 **	64.5%
Brain, spinal cord, nervous system	64	42.84, 1.23	46.73, 1.23	4.16, 1.02 †	0.003 **	48.1%

Pre- to post-rehabilitation within-group change, *p* < 0.05. ** Pre- to post-rehabilitation within-group change, *p* < 0.01. † Estimated marginal (EM) mean change exceeded the minimal important change (MIC; 2 points) [42]. EM means adjusted for the following covariates: age, sex, rehabilitation service type (PT or OT), visits, and length of care.

## Data Availability

Data available by request is subject to approval.

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
