# Peer review of "Impact of Real-World Outpatient Cancer Rehabilitation Services on Health-Related Quality of Life of Cancer Survivors across 12 Diagnosis Types in the United States"

_cancers, 2024, doi:10.3390/cancers16101927_

Round 1

Reviewer 1 Report

Comments and Suggestions for Authors

Overall, this study provides real world data on the impact of PT and OT services for a wide variety of cancer patients. It uses an accessible, standardized metric of function to describe the outcomes of a limited set of rehabilitation services to measure the outcomes in a large sample from 21 states.

One major concern is that the authors frame their manuscript around the idea that participation in rehabilitation improves QOL outcomes. However, the data presented comes only from patients who filled out the PROMIS measure on entry to rehabilitation and again at discharge from rehabilitation. The authors do not present data to support that the vast majority of patient that were seen over this timeframe both completed the baseline measure AND returned to their therapy visits until being discharged from treatment, filling out the outcome again at the end of therapy. Thus, I would suggest that at a minimum, the authors report on the number of patients that were seen for 2 or more visits over that same time and to frame their discussion of the impact of rehabilitation for those that complete therapy throughout the manuscript.

Another issue, that may just be a limitation due to the difficulty abstracting this data, is a lack of understanding if patients were receiving concurrent medical treatment or were all post-oncologic treatment during their rehabilitation. Understandably, this is a messy issue, but even if a significant number of the patients were continuing to receive oncologic treatments such as chemotherapy or radiation, then the improvements in function are especially noteworthy.

I appreciate that this manuscript is reporting on population means/medians. However, it would be interesting to know what % of each cancer population met the minimally important metric for each measure. In this way, the data could be used exactly as the authors state, to encourage referral and completion of rehabilitation. For example, if a therapist could state that 95% of head and neck patients made clinically important improvement in their physical function if they completed therapy, this may assist with completion of treatments.

Minor comments:

Abstract and Results: Make sure that decimals are appropriate for the outcome reported (ie. Age mean of 66.0 if age was recorded in whole years only)

Introduction:

Line 66: What type of trial are your specifically referring to here? Medical trails, rehabilitation trials, or both? Also see this issue in the discussion.

Methods:

Case characteristics: Missing cancer diagnosis in list. I am assuming that the ICD-10 codes were used to establish the cancer type. Could make this clearer. Were patients receiving concurrent medical treatment or were they all post-oncologic treatment. If it is not able to be pulled from the medical records, this would be a point of discussion, as it could impact the degree of change that would be expected.

Lines 106-107: Reporting on the minimally importance difference. Please make it clearer that you used the lowest suggested value in your reporting.

Results:

Were no patients seen by both PT and OT? It would seem that some may have seen both.

Discussed above: Over this same time period, how many patient had initial evaluations and PROMIS scores but either discontinued treatment or did not fill-out the PROMIS at the end of treatment? This speaks to the representative nature of the sample population.

Discussed above: Line 146: While it is true that on average all cancer types met the minimally improved metric, it would be important to know what percent of patients for each cancer type met the clinically improved metric.

Line 148: You state “Of those who did not meet the MIC…” – I am assuming that you are referring to cancer not patients.

Discussion:

Discussed above: Line 202: Not just participation in, but completion of rehabilitation..

Line 287: “seen at a single institution”… this is a large network, so you may be selling yourself short here. Maybe state “seen within a large rehabilitation network…”

In line 287, it stated PT and/or OT services, however Table 1 seems to split them as one or the other. This data or statement needs to be clarified and/or corrected.

Comments on the Quality of English Language

Generally well written. Clearly, proofing of the typeset needs to occur.

Author Response

Thank you all for your thorough review and comments. We have edited our manuscript (all changes in blue text) and have provided a point by point response. Thank you for your consideration of this revised paper. 

Reviewer 2 Report

Comments and Suggestions for Authors

The authors are interested in understanding whether cancer patients/survivors experience poorer HRQOL.

The sample size that was considered in this prospective study was great - diverse in terms of sex and ethnicity. Although the patients are largely on federally funded insurance ( which may indicate that it is skewed to a particular sociodemographic).

1. Is the difference in the parameters that were considered in Figure 1 significant?

2.Are stages of cancers identical for all the patients considered? If not that may be another parameter to look at ...

3. Were other health behaviours such as alcohol use, tobacco , physical activity that could affect all the parameters mentioned in the paper considered?

Reviewer 3 Report

Comments and Suggestions for Authors

minimum changes required 

see attached manuscript

great work 

but I would like to request that you add that your study was USA based and cannot be translated in other countries where health care is different for example in Canada

also OT was not used a lot

so why don't you add a limitation and question why OT was not utilized as much as PT

this part should be in your discussion because it is flagrant that the utilization was minimum

Reviewer 4 Report

Comments and Suggestions for Authors

Authors have written a manuscript regarding the impact of real-world outpatient cancer rehabilitation services on health-related QoL of cancer survivors across different diagnosis types. This manuscript is of interest, as includes patients with many different characteristics but treated with real-world physiotherapy or occupational therapy. However, some changes have to be made before publishing this study. 

In the introduction, it is not really clear the gap in the evidence regarding QoL in cancer patients. Please clarify this information before explaning your outomes. I would also recommend the authors to include an hypothese about the results they may obtain with their research. 

It is really interesting that you include so many different types of cancer diagnosis; but there are some differences between the n of head and neck cancer patients and the n of leukemia patients. This difference should be mention on your discussion and limitations, as it could be possible that your results are due to the differences on the sample size of each cancer diagnosis. 

When doing these changes, in my opinion this manuscript is of interest to be published. 

Round 2

Reviewer 1 Report

Comments and Suggestions for Authors

The authors have appropriately addressed all reviewer concerns.

Reviewer 2 Report

Comments and Suggestions for Authors

The authors have addressed my concerns either in the results or discussion.